

# Cloud type classification using deep learning with cloud images

Mehmet Guzel[1], Muruvvet Kalkan[1], Erkan Bostanci[1], Koray Acici[2] and Tunc Asuroglu[3]

[1] Department of Computer Engineering, Ankara University, Ankara, Turkey
[2] Department of Artificial Intelligence and Data Engineering, Ankara University, Ankara, Turkey
[3] Faculty of Medicine and Health Technology, Tampere University, Tampere, Finland

## ABSTRACT

Clouds play a pivotal role in determining the weather, impacting the daily lives of everyone. The cloud type can offer insights into whether the weather will be sunny or rainy and even serve as a warning for severe and stormy conditions. Classified into ten distinct classes, clouds provide valuable information about both typical and exceptional weather patterns, whether they are short or long-term in nature. This study aims to anticipate cloud formations and classify them based on their shapes and colors, allowing for preemptive measures against potentially hazardous situations. To address this challenge, a solution is proposed using image processing and deep learning technologies to classify cloud images. Several models, including MobileNet V2, Inception V3, EfficientNetV2L, VGG-16, Xception, ConvNeXtSmall, and ResNet-152 V2, were employed for the classification computations. Among them, Xception yielded the best outcome with an impressive accuracy of 97.66%. By integrating artificial intelligence technologies that can accurately detect and classify cloud types into weather forecasting systems, significant improvements in forecast accuracy can be achieved. This research presents an innovative approach to studying clouds, harnessing the power of image processing and deep learning. The ability to classify clouds based on their visual characteristics opens new avenues for enhanced weather prediction and preparedness, ultimately contributing to the overall accuracy and reliability of weather forecasts.

## INTRODUCTION

The field of meteorology, which has gained widespread recognition across the globe in recent years, encompasses a multitude of factors when formulating weather forecasts. Among these factors, humidity, temperature, and pressure stand out as crucial elements. However, it is the study of clouds that lies at the heart of weather prediction. Meteorological systems demonstrate remarkable accuracy, particularly in short-term forecasts, achieved through the meticulous classification of clouds based on their coverage percentage and types (*Kalkan et al., 2022*).

In addition, in climate change research clouds have a key role. While clouds affect the climate, at the same time climate also influences clouds, in other words there is a powerful correlation between the two. Although this correlation is complex and uncertain at times,

Corresponding author
Tunc Asuroglu,
tunc.asuroglu@tuni.fi

**Table 1 Cloud type heights.**

| Cloud type | Low clouds | Mid-height clouds | High clouds |
|---|---|---|---|
| Altocumulus (Ac) | | X | |
| Altostratus (As) | | X | |
| Cumulonimbus (Cb) | X | | |
| Cirrocumulus (Cc) | | | X |
| Cirrus (Ci) | | | X |
| Cirrostratus (Cs) | | | X |
| Cumulus (Cu) | X | | |
| Nimbostratus (Ns) | | X | |
| Stratocumulus (Sc) | X | | |
| Stratus (St) | X | | |

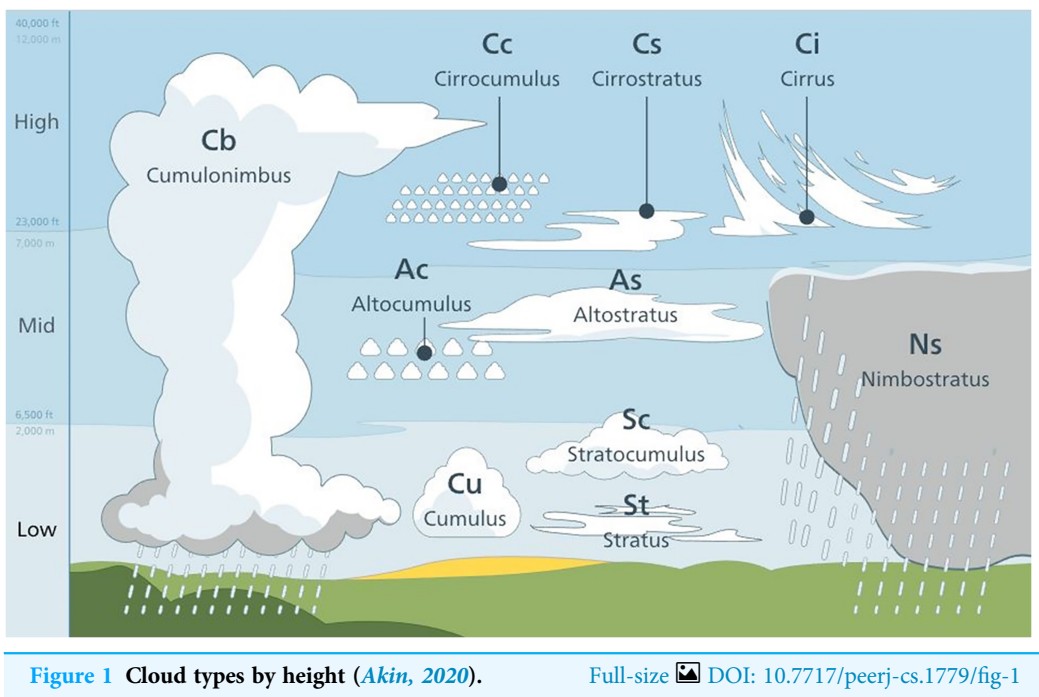

**Figure 1 Cloud types by height (*Akin, 2020*).**

two important examples could be clouds' effect on water resources distribution and water cycle (*World Meteorological Organization, 2017*). Thus, it should be considered that identifying and understanding cloud types plays a critical part not only in determining weather conditions but also in fathoming climate change better.

Clouds are fundamentally grouped under ten categories. They are classified by two factors: the first is the height and the second is the shape of clouds. According to their height, the clouds are grouped under three superclasses, see Table 1 and Fig. 1.

**High clouds:** cirrus, cirrocumulus, cirrostratus.

**Mid-height clouds:** altocumulus, altostratus, nimbostratus.

**Low clouds:** stratus, stratocumulus, cumulus, cumulonimbus.

Each cloud type carries distinct characteristics, allowing for specific interpretations. Consider the following examples: Stratocumulus clouds often result in a light drizzle, while Nimbostratus clouds bring forth heavy rainfall accompanied by thunder and hail. The presence of Altostratus clouds indicates the likelihood of extreme weather events, as they have the potential to develop into nimbostratus clouds, which can generate rain as well as snow (*Akin, 2020*). Cirrus clouds, although capable of producing precipitation, never reach the ground as they evaporate before reaching the surface. These examples highlight how the shape and altitude of clouds enable experts to make precise weather predictions. Leveraging modern artificial intelligence technologies, rather than relying on outdated methods, can significantly reduce prediction errors.

Deep learning, an advanced field within artificial intelligence, offers a practical technique for image classification in conjunction with image processing. It operates through specialized architectures known as artificial neural networks, which mimic the structure and functionality of human and animal brains (*Alzubaidi et al., 2021*). Taking this concept further, convolutional neural networks (CNNs) perform calculations using artificial neurons that traverse specific layers. As a powerful branch of deep learning, CNNs are particularly effective in image classification tasks, hence their selection for this experiment (*Yılmaz et al., 2020*).

In everyday applications of deep learning, a method called transfer learning proves highly valuable. Transfer learning involves leveraging previously acquired knowledge when encountering new problems. By retaining and reusing learned information, this approach enables easier problem-solving and faster results. Unlike traditional machine learning methods that require relearning for each task, deep learning methods with transfer learning achieve higher accuracy rates by leveraging results and weights acquired from previous tasks. Therefore, the research utilized the transfer learning method in training the deep learning models.

The academic work is elaborated in the following sections. The Literature Review section presents relevant prior studies and the literature on the employed methods. The Methodology section explains the chosen experimental approach, including the input dataset and a step-by-step algorithm. The Experimental Results and Discussion sections provide the outcomes of the experimental program and discuss them, respectively. Finally, the Conclusion section summarizes the study, offers insights into the future, and provides suggestions for further improvement.

## LITERATURE REVIEW

In recent decades, application of artificial intelligence technologies have become much more common compared to before. Although various fields employ AI solutions for certain problems, there are not many studies and applications on the issue of "classification of cloud types". With neural networks, the task of classifying cloud images had been done with approaches that contain a few linear layers. Then, neural networks with nonlinear and non-parametric layers became able to handle it. These architectures tried to base their work on cloud textures as distinguishing features (*Lee et al., 1990*). Later on, the cloud pictures from satellites were classified with extracting texture features *via* Singular Value

Decomposition (SVD) and Wavelet Packet (WP) matrix transforms (*Tian et al., 1999*; *Azimi-Sadjadi & Zekavat, 2000*). *Heinle, Macke & Srivastav (2010)* managed to classify cloud types from sky images, which is the same task this study aims to deliver, but with a different approach, and it follows a K nearest neighbor based algorithm. Another study with the same type of input and goal used the scale invariant feature transform (SIFT) and a linear support vector machine (SVM) to accomplish the task (*Xiao et al., 2016*). In addition, academic works that preferred semantic approaches also undertook the classification of cloud types (*Liu et al., 2019*; *Zhang et al., 2020*). The same dataset of the experiment was processed for classifying weather's being cloudy or clear and calculating the cloud coverage percentage in a study (*Kalkan et al., 2022*). However, the main inspiration of this work is the CloudNet designed by *Zhang et al. (2018)*, using the same dataset, the Cirrus Cumulus Stratus Nimbus (CCSN) Database, with the same goal and following different but similar methods involving CNNs. There are also other research teams who experimented with CCSN dataset. One of them are *Gyasi & Swarnalatha (2023)* who designed a CNN model that is composed of modified MobileNet blocks and their model is called Cloud-MobileNet. A new channel attention module is developed by *Zhu et al. (2022)* to classify CCSN dataset. Lastly, a transfomer model approach is preferred by *Li et al. (2022)* in their study of cloud classification in CCSN.

The target of this study is to experiment with CNNs based on pre-trained models (implementing transfer learning) and to improve the outcomes produced. With the techniques applied correctly, it is expected that the proposed approach of this academic work surpasses the preceding studies.

A deep learning method called transfer learning is using the information obtained while solving a past problem when an entirely different problem is encountered, which is similar to how humans solve problems. Keeping previously learned information and using it for new situations allows both to produce easier solutions and to get results in a shorter time. While traditional machine learning methods relearn for each new task, deep learning architectures using transfer learning yield higher accuracy rates, because they can benefit from previously learnt results or weights. Transfer learning allows working on datasets from different fields (*Pan & Yang, 2009*). For all these reasons, the transfer learning was chosen for experimental models while conducting the research.

In the field of deep learning, various techniques are employed to enhance the effectiveness of models. One such technique, known as freeze-out fine-tuning, aims to elevate accuracy rates by selectively disabling (or freezing) certain layers within a model during training. While the untouched layers are trained conventionally, the weights and biases of the frozen layers remain unchanged (*Brock et al., 2017*). In this study, fine-tuning was implemented by freezing one third of the model during the training process. The experimental results section provides a comprehensive analysis of the impact and outcomes derived from the fine-tuning procedure.

Since the method proposed by the experiment is transfer learning, certain pre-trained models are selected as bases for main models: MobileNet V2, Inception V3, EffcentNetV2L, VGG-16, Xception, ConvNeXtSmall and ResNet-152 V2. MobileNets,

mobile applications as their major target, were first proposed by Google researchers and they strive to be fast, lightweight and adaptable (*Howard et al., 2017*, *2018*).

Built by Google engineers and researchers, Inception architectures have dedicated layer blocks called inception blocks. These blocks have a genuine parallel convolutional layer structure, and they contain a concat layer at the end which concatenates the outputs of the predecessor parallel layers into a single output. Along with the unorthodox parallel convolutional layers, Inception models settle the setbacks of having a very deep network by calculating loss values in the intermediate layers and combining them with the final loss value (*Christian et al., 2015*).

Published by Google, EfficientNets are powerful models that are based on a MobileNet like structure and scaled efficiently into much more accurate. Scaling CNNs is a common practice and it can be done on width, depth or resolution of layers, but finding optimal scaling coefficients was problematic. What EfficientNets proposed is to do a compound scaling on an existing architecture following a certain way. The baseline model for compound scaling has an architecture similar to MobileNet V2 and the compound scaled EfficientNets have produced superior results compared to MobileNets and ResNets on ImageNet dataset (*Tan & Le, 2019*, *2021*).

Developed by the Visual Geometry Group (VGG) of Oxford University, VGG models are deep architectures consisting of small convolutional filters. VGG models achieve their goal of high accuracy, which is proven by rankings of ILSVRC-2014 (ImageNet Large Scale Visual Recognition Competition) (*Simonyan & Zisserman, 2014*).

Developed by *Chollet (2016)* from Google, Xception is an architecture that takes the structure of the Inception models to extreme, so Xception model managed to surpass Inception models and it is named "Extreme Inception".

After the introduction of transformers, CNNs started to go out of date. To catch up with transformers certain design approaches are taken to modernize CNNs, so ConvNeXts were proposed (*Liu et al., 2022*).

First designed and implemented by Microsoft researchers, ResNets aspire to build deeper layered architectures and resolve the shortcomings of having deep layers by residual mapping (*He et al., 2015*, *2016*).

## METHODOLOGY

One of the most common methods used on image classification today is convolutional neural networks. With the aim of classifying cloud types by their pictures, CNNs are chosen for this academic work. In another previous study, the same classification has been made on the same dataset with an architecture called CloudNet, which prefers this technology and builds its own model from scratch. However, in this experiment, pre-trained models of deep learning technology were preferred with the transfer learning technique.

The dataset used in the experiment consists of 11 folders of pictures, each representing a cloud type (*Liu, 2019*). Clouds are basically divided into ten classes. The eleventh type, contrail (Ct) is the vapor trail of airplanes as a separate class, since they look similar to actual clouds from ground. The dataset consists of 2,543 images. A total of 70% of them are

**Table 2 An example partition of cloud classes into training validation and testing classes.**

| Cloud type | Train | Validation | Test |
|---|---|---|---|
| Ac | 154 | 45 | 22 |
| As | 131 | 32 | 25 |
| Cb | 174 | 50 | 18 |
| Cc | 190 | 56 | 22 |
| Ci | 99 | 30 | 10 |
| Cs | 197 | 54 | 36 |
| Ct | 140 | 44 | 16 |
| Cu | 127 | 31 | 24 |
| Ns | 191 | 54 | 29 |
| Sc | 240 | 70 | 30 |
| St | 141 | 37 | 24 |

**Table 3 Hyperparameters of the experiment.**

| Hyperparameter | Value(s) |
|---|---|
| Train, validation, test split ratios | 0.7: 0.2: 0.1 |
| Batch size | 32 |
| Dropout | 0.2 |
| Learning rate | 0.0001 |
| Optimizer | Adam |
| Loss type | Categorical cross entropy |
| Epoch count | 100 |
| Fine tuned layers | First 1/3 of base model layers are frozen out |

reserved for the training dataset, 20% for the validation dataset and the remaining 10% for the test dataset. This split is random but balanced and made for each pre-trained model's experiment. All mentioned dataset partitions by their classes are given in the Table 2.

The Google Colab environment used in the study was preferred because it meets technical requirements of this experiment, provides free access, does not require any installation and it is cloud-based and suitable for collaboration. It is an environment where machine learning and deep learning projects are often implemented. It is compatible with the Python programming language and Tensorflow, which is an open source library containing CNN structures and functionalities.

The hyperparameters chosen for this experiment are given in the Table 3.

The experimental program follows certain steps, these steps start with fetching the input dataset. It continues with building a model using the determined pre-trained model as base and training it. Finally the experiment steps end with evaluating the final model by test dataset and producing predictions on some samples. This procedure is repeated for each seven pre-trained models, only changing the base model, so that scientifically appropriate

outcomes would be produced and interpreted. The experimental program is described in the following steps:

1. The images are loaded, then rescaled to the size of $224 \times 224$.

2. The dataset consisting of 2,543 images is partitioned into 70% as training, 20% as validation and the remaining 10% as test dataset.

3. Data augmentation layer is defined. With "data augmentation", the number of images is increased synthetically. The image set with 2,543 images would become more than tenfold of itself.

4. Each image in the dataset consists of pixels that are 1D matrices of RGB (red, green, blue) values. These RGB values are in the range of [0,255]. These values are scaled into the range [0,1] or [−1,1] to speed up and simplify the calculations. The preprocess layers of the pre-trained models decide which target interval will be used. Therefore, the required preprocess layer would be defined.

5. Pre-trained models, named as base models in this study, are loaded into the program from Tensorflow library.

6. Models have multiple layers. During the training, the layers at a certain rate are frozen and the training is done with the remaining layers. Aforementioned, this process is called freeze-out fine tuning technique. In the experiment, training of about the first one-third of the layers are disabled, namely frozen.

7. The pooling layer is created. In this layer, the global average pooling method is preferred.

8. Then, the prediction layer which will produce outputs of the model is defined. Softmax is chosen as the activation function, because input images' probabilities of belonging to each class are desired and softmax is the most fit solution to this issue.

9. The main model is built with these layers and base model. All pre-trained models become the base model for its own experiment and the same steps are repeated for each seven of them. Layer structure of main models are given in Fig. 2.

10. Next, the main model is compiled with categorical cross entropy loss and evaluation metrics. The evaluation metrics are accuracy, precision, recall, F1-score, ROC AUC (Area Under the Curve of Receiver Operating Characteristic).

11. In the following step, with 100 epochs, training of the main model is initialized. Train and validation datasets are processed during the training. In each epoch, loss and metric results are saved into a training history record.

12. After training is finished, recorded history of metric/loss results by epochs are displayed as learning curves, since such data is perfect for visualizing the learning process of CNNs.

13. In order to determine the success of an experimental main model, the trained main model is evaluated by the test dataset, again with the same loss and metrics. These final results allow main models to be compared.

Inputs

Data Augmentation Layer

Preprocess Layer

Pre-Trained Base Model

Pooling Layer

Dropout Layer

Prediction Layer

Outputs

**Figure 2  Layer structure of main models.**     

14. Finally, a sample of predictions made on input images are displayed with the images and their graphical prediction results. Some of them can be seen in Fig. 3.

These steps are also given in the following Algorithm 1.

## EXPERIMENTAL RESULTS

All experiments were done in the cloud environment of Google Colab and the hardware used is provided by it, as well. The GPU that ran the program is NVIDIA Tesla T4 whose specifications include a memory with the size of 16 GB GDDR6 and the bandwidth up to 320 GB per second.

Aforementioned, the trained models make some predictions on random sample inputs from the test dataset. Test dataset is chosen for the sample input, because the models were not exposed to the test dataset, so the trained models are unbiased towards the test dataset
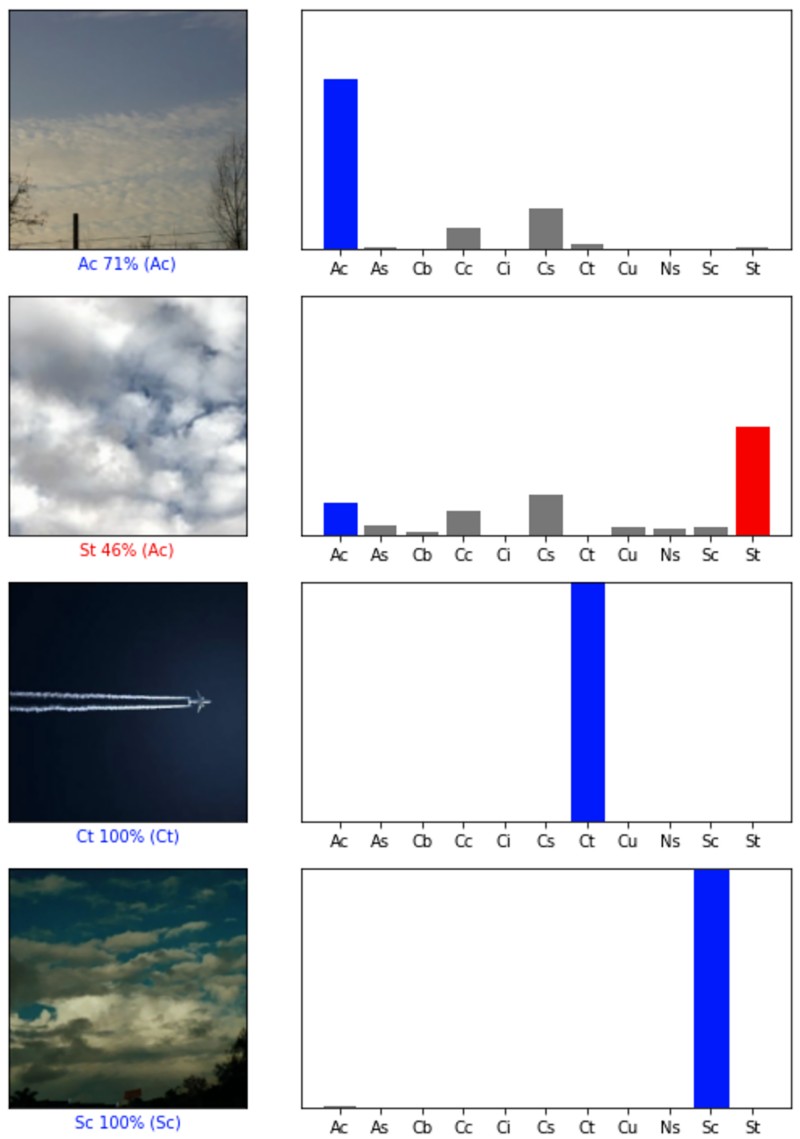

**Figure 3 Sample prediction results (*Liu, 2019*).** Images that are predicted correctly are blue labeled, while incorrect ones are red labeled. In the bar chart, probabilities of belonging to each class calculated by a model are given. Real class is displayed as blue. If the highest probability is wrong, then it is red. The rest of the probabilities are dyed to gray.

and would produce more objective results. The type of a prediction is in the form of a 1D array/vector, because the final activation function of the prediction layer in all main models is softmax function and classification type is categorical. The length of the array is the number of classes, here it is eleven. Each element of the array is a number between zero and one ([0, 1]) and the sum of all elements is always one. Namely, each index represents a class and the element of that index is the input image's probability of belonging to that class. The final single predicted class is the one with the highest probability. Some samples of these prediction results are displayed in the Fig. 3. Bar charts display the mentioned array of probabilities, labels with percentages represent the final predicted classes along

**Algorithm 1 Building the experimental model, model training with fine tuning and evaluation.**

Fetch the dataset

Split them as training, test and validation datasets

Approximately 70% training, 20% validation, 10% test

$baseModel \leftarrow$ get the pretrained model (MobileNet V2, VGG-16, ResNet-152 V2, Xception, Inception V3, EfficentNetV2L, ConvNeXtSmall)

$model \leftarrow$ create the experimental model

$model.layers \leftarrow$ empty, initial value

$model.layers.insert(inputLayer)$

$model.layers.insert(dataAugmentationLayer$

$model.layers.insert(preprocessLayer)$

$model.layers.insert(baseModel)$

$model.layers.insert(globalAveragePoolingLayer)$

$model.layers.insert(predictionLayer)$

$baseModel.trainable \leftarrow True$

$fineTuneAt \leftarrow$ approximately 1/3 of the number of layers in the model

**for** $k \in \{0, \ldots, fineTuneAt\}$ **do**

$baseModel.layers[k].trainable \leftarrow False$

**end for**

$metrics \leftarrow [accuracy, loss, precision,$

$recall, f1Score, roc]$

$model.compile(metrics)$

Train the $model$

Draw the learning curves

Evaluate $model$ by the test dataset

with their probability and finally the actual classes they belong to are displayed within parenthesis. The color blue means accurate guesses, the gray demonstrates the other probabilities and the red shows the incorrect predictions.

## Evaluation metrics

For each model, training and test scores by the six metrics, accuracy, loss, precision, recall, F1-score, ROC AUC, are produced and saved for examination and comparisons. In addition, training runtimes are also taken into consideration for each model in order to

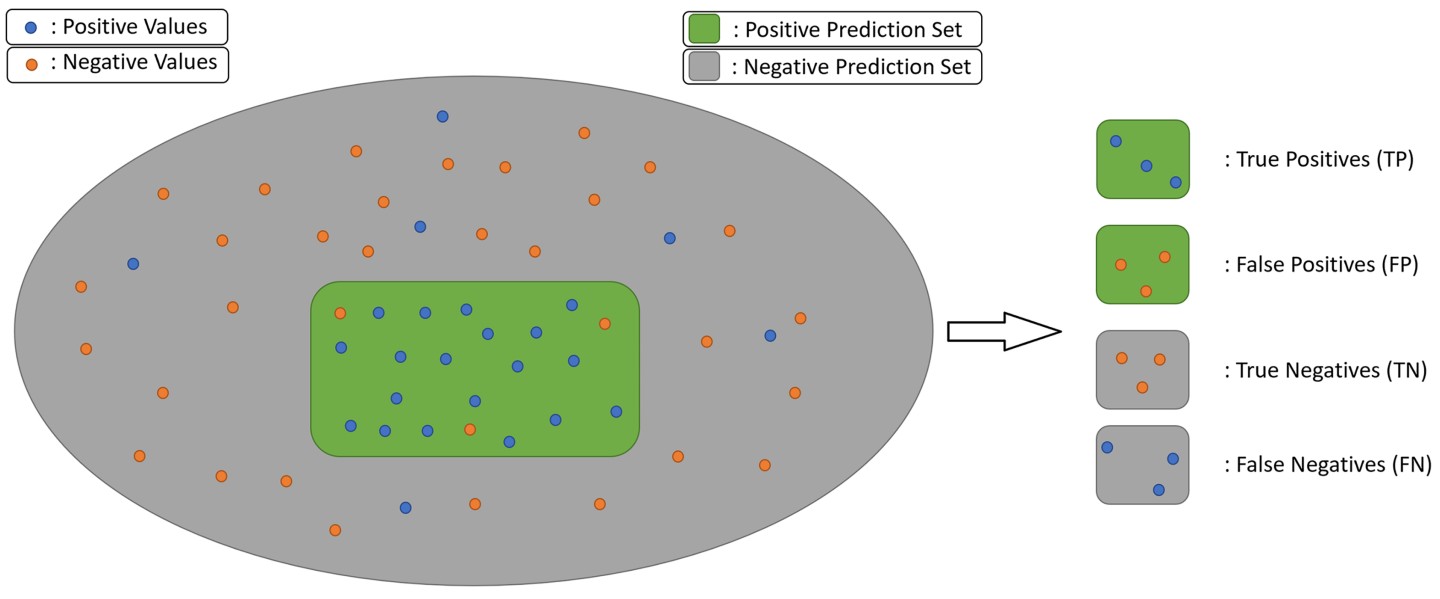

**Figure 4 Prediction value definitions.**

measure the time performance. All these factors explicitly and numerically are given in this section and fully scrutinized in the following Discussion section.

Each of the six metrics are calculated for all the eleven classes and the arithmetic average of per class scores is calculated and accepted as the overall score. Except for the loss, the five metrics are calculated from prediction values, the number of true/false positives and negatives, Fig. 4.

- **TP:** Number of correctly predicted positive values, *true positives*
- **TN:** Number of correctly predicted negative values, *true negatives*
- **FP:** Number of incorrectly predicted as positive, but actually negative values, *false positives*
- **FN:** Number of incorrectly predicted negative, but actually positive values, *false negatives*

Accuracy is the most commonly accepted metric for evaluating a model's overall success, Eq. (1).

$$Accuracy = \frac{TP + TN}{TP + FP + TN + FN} \tag{1}$$

Precision represents a prediction system's skill of not hitting false positives. The higher it is, the purer the set of predicted positives, in terms of true positives, Eq. (2).

$$Precision = \frac{TP}{TP + FP} \tag{2}$$

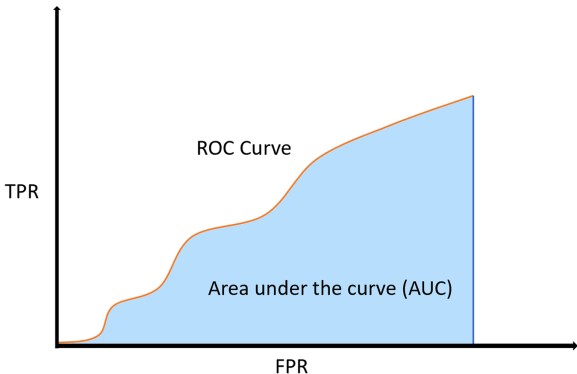

**Figure 5  ROC curve and its AUC.**               

Recall shows how good the system is at not missing possible positives. The higher it is, the more actual positives are predicted as positives, Eq. (3).

$$Recall = \frac{TP}{TP + FN} \tag{3}$$

Precision and recall are opposite polars in prediction systems. Depending on the domain and the goal of the system, one of them may become more important, but in this academic work a balance of them is preferable. The metric chosen for evaluating the balance of precision-recall is the F1-score. The higher it is, the more quality predictions are produced by the model. The F1-score formula is given in Eq. (4).

$$F_1 = \frac{TP}{TP + \frac{1}{2}(FP + FN)} \tag{4}$$

If a model's skill on predicting correctly *vs* incorrectly wanted to be presented, then the receiver operating characteristic (ROC) curve could easily show it. Axis of the ROC curve are TPR and FPR, see Eqs. (5) and (6). However, the numeric metric which shows the skillfulness of a model is actually the area under the curve (AUC) of ROC, or shortly ROC AUC, see Fig. 5.

$$TPR = \frac{TP}{TP + FN} \tag{5}$$

$$FPR = \frac{FP}{FP + FN} \tag{6}$$

The final metric is the cross entropy loss. There are various loss types, but the categorical cross entropy loss is selected. The loss represents how far is the prediction from the actual value. The less it is, the more accurate predictions are being made by the architecture.

Since the training is done with epochs and in each epoch evaluation metrics are calculated, it is possible to draw learning curves. Also, the models are trained with two datasets, training and validation, so for each model and accuracy-loss metrics there are two learning curves.

First of all, for all the models and two datasets, accuracy and loss curves are much more varying depending on the model. The following observations are made on the learning curves. MobileNet V2 based model's learning curves display that this model learning and mastering process is overall moderate in Figs. 6 and 7. The curves of the VGG-16 are gradual and even close to linear in all metrics in Figs. 8 and 9. The learning curves of the ResNet-152 V2, Inception V3, EfficientV2L, Xception, ConvNeXtSmall based models are all steep, see Figs. 10–19.

In addition, a confusion matrix is made for each model. Since this is a multi-class classification, the confusion matrices are based on actual *vs* predicted classes of test dataset images. Since dataset's random split of train, validation and test is made for each model, the image classes are distributed differently. The confusion matrices by classes are given in Figs. 20 and 21.

The final evaluation was done with the test dataset, so by each metric models based on transfer learning can be objectively compared. By accuracy and loss, Xception gave the best result with 97.66% and 0.06, followed closely by, and equal in loss, ConvNeXtSmall. Xception also got all the highest precision, recall and F1-Score by wide margins. In terms of ROC AUC, ConvNeXtSmall with the first place got 99.99% and Xception with the second got a close 99.98%. Overall Xception got all the best scores, except for ROC AUC but became second with a 0.01% difference. See Table 4 along with Figs. 22–24.

In order to test the affect of freezeout fine tuning on the proposed models, the same experiment is repeated without it. Namely, all layers of the base models are enabled for training. The accuracy results of them are compared and given in Table 5. As it can be seen from these results, freezeout fine tuning technique has an overall positive impact on the accuracy of all models.

The best produced accuracy of the proposed method with Xception based model is compared against some of the latest literature that also scored themselves with CCSN dataset. The comparison is given in Table 6 and it can be seen that our proposed approach cam best among the other DL models.

For machine learning solutions one of the biggest challenges is time performance of models/algorithms. Time efficiency of algorithms depending on the input size n can be expressed with the big O notation like O(n). However, for complex solutions like CNN architectures of deep learning, the time efficiency is mostly independent of input size. The impacting factors of their performance are the model's structural features, for example the number, width and depth of layers, and they are not single numbers that can be shown like the n of the O(n). Thus, it is not possible to express the time efficiency of CNNs with big O notations. Although formulating the time performance is not possible, the runtime speed of the training sessions of CNNs can be measured in seconds, so that model speed can be compared with each other.

Since it is built for mobile purposes with an expectancy of being fast, MobileNet V2 became the fastest model of all by a wide margin. Inception V3, Xception, VGG-16, ResNet-152 V2 had moderate time efficiency, while ConvNeXtSmall and EfficientNetV2L got the last places. See Table 7.

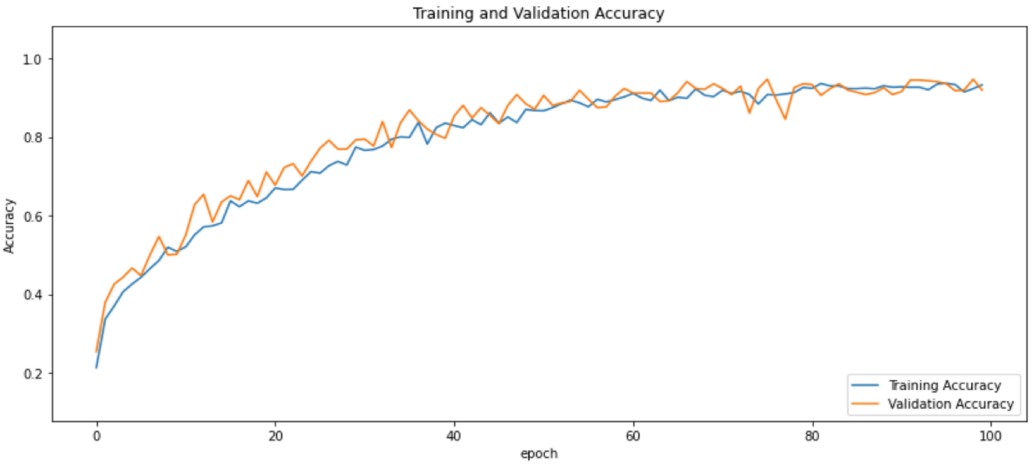

**Figure 6 Accuracy learning curves of MobileNet V2.**

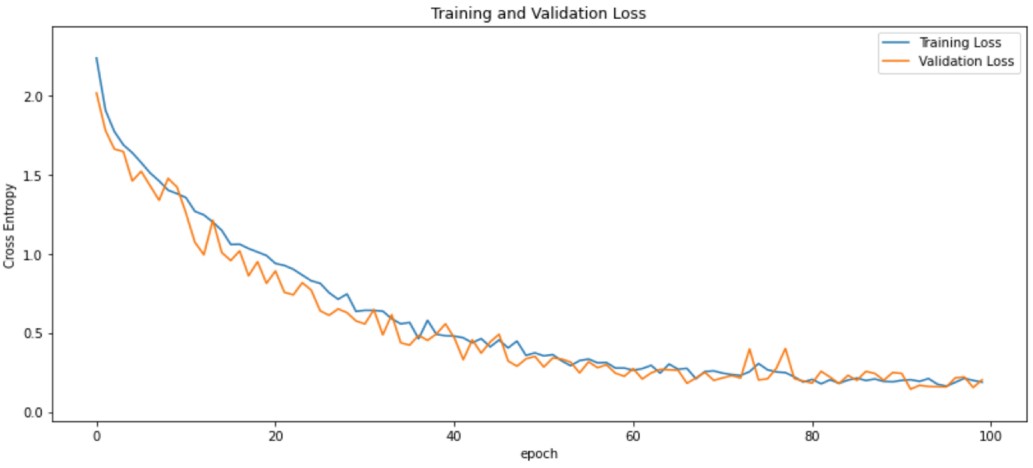

**Figure 7 Loss learning curves of MobileNet V2.**

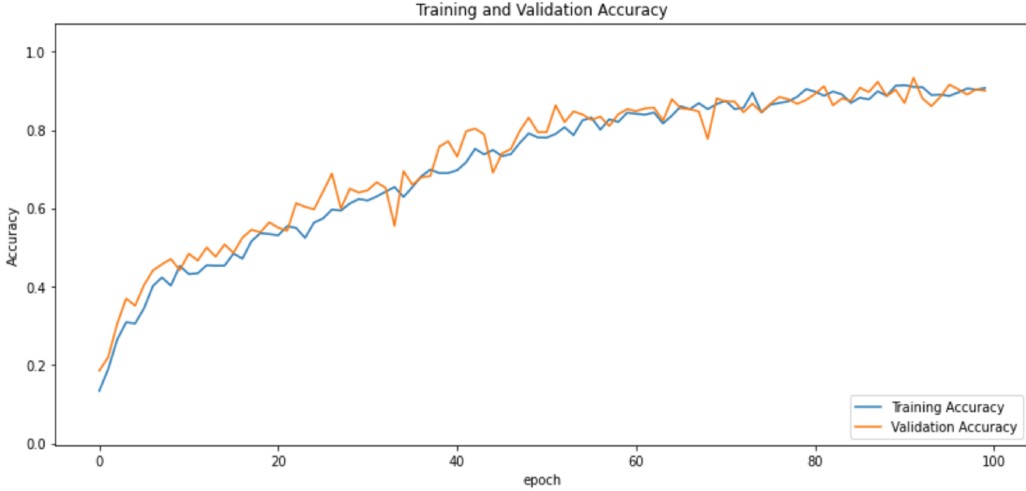

**Figure 8 Accuracy learning curves of VGG-16.**

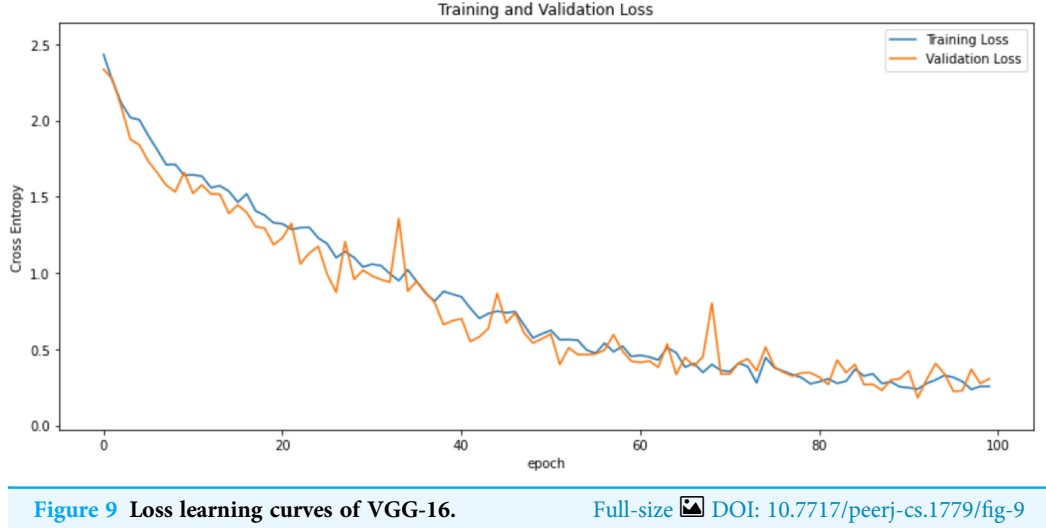

**Figure 9 Loss learning curves of VGG-16.**

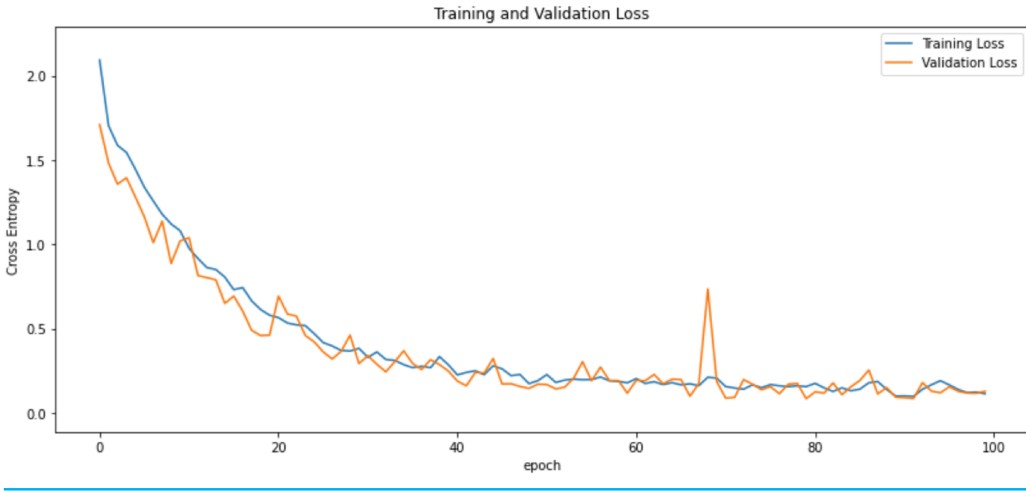

**Figure 10 Accuracy learning curves of ResNet-152 V2.**

**Figure 11 Loss learning curves of ResNet-152 V2.**

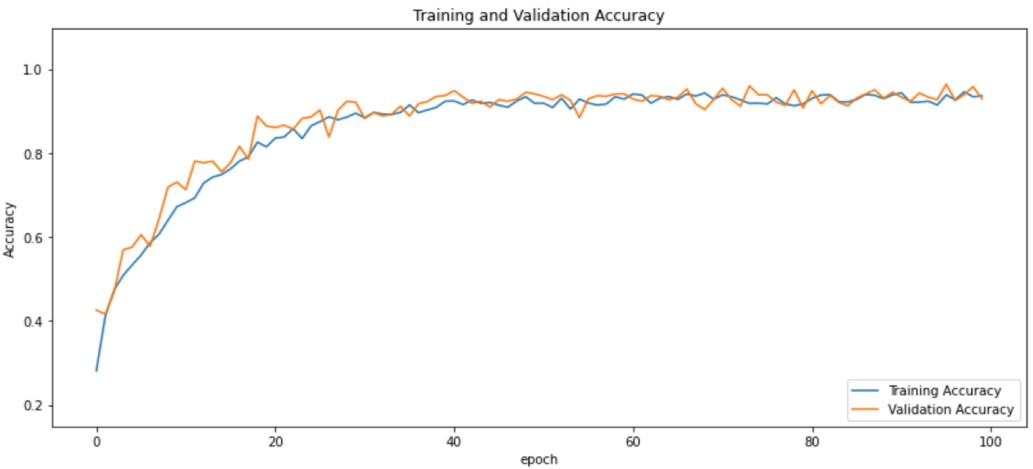

**Figure 12  Accuracy learning curves of Inception V3.**

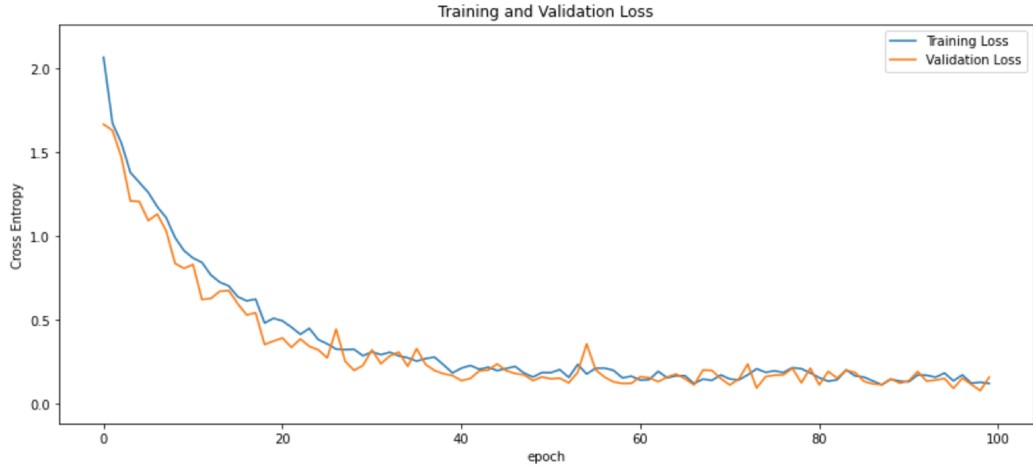

**Figure 13  Loss learning curves of Inception V3.**

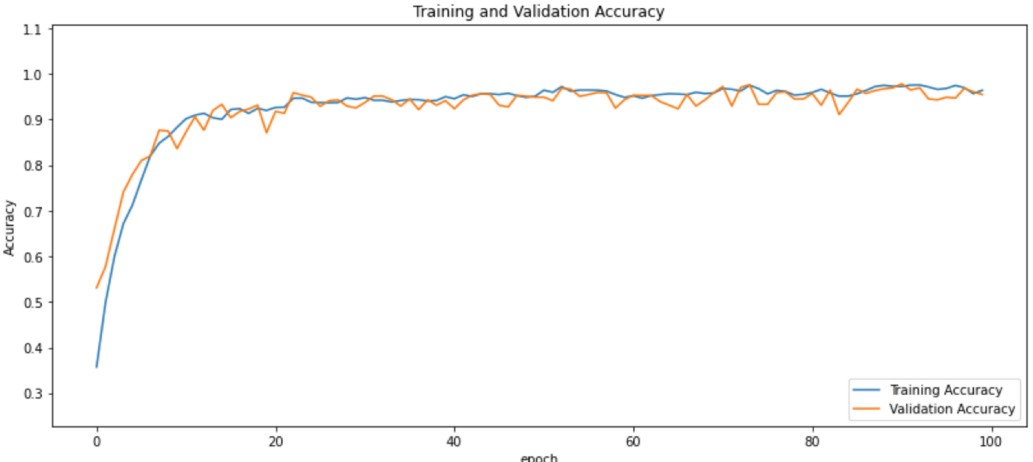

**Figure 14  Accuracy learning curves of EfficentNetV2L.**

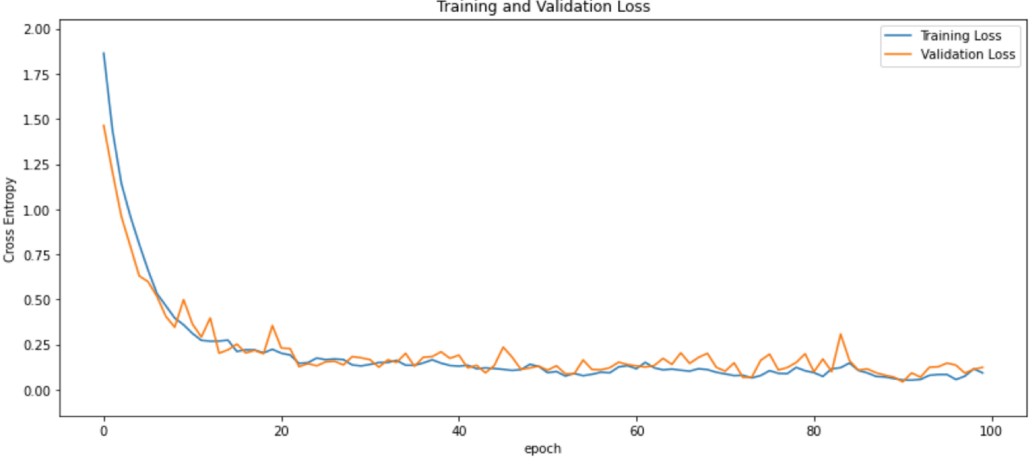

**Figure 15 Loss learning curves of EfficentNetV2L.**

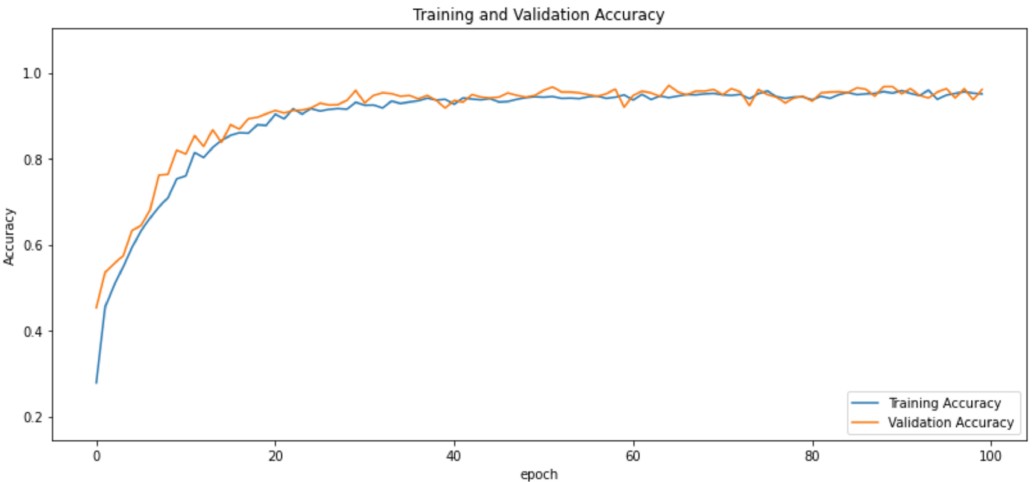

**Figure 16 Accuracy learning curves of Xception.**

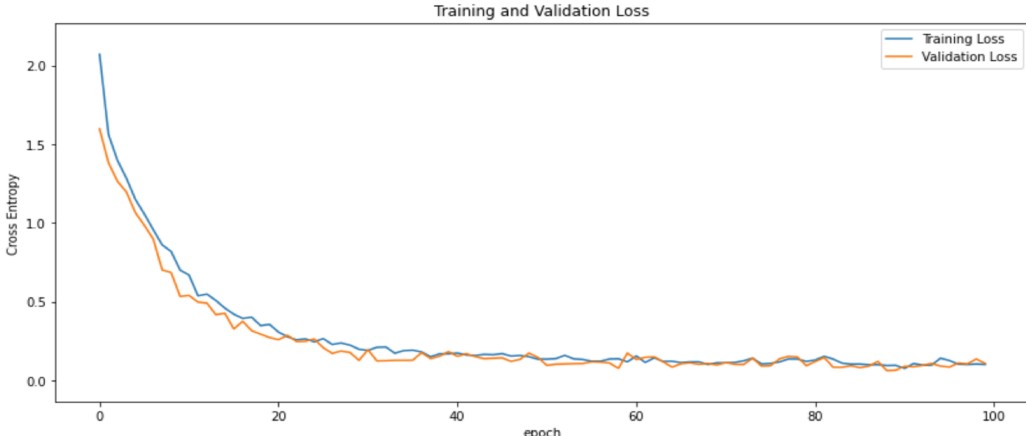

**Figure 17 Loss learning curves of Xception.**

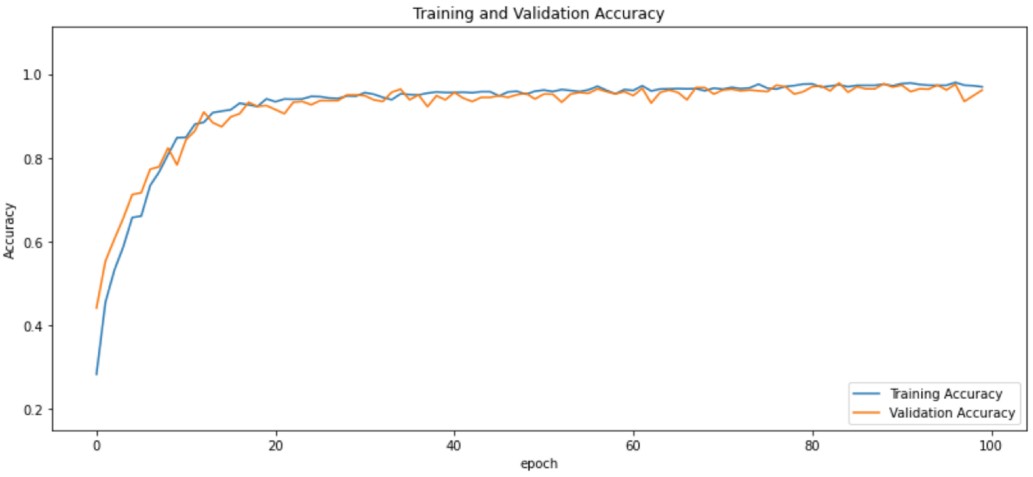

**Figure 18  Accuracy learning curves of ConvNeXtSmall.**

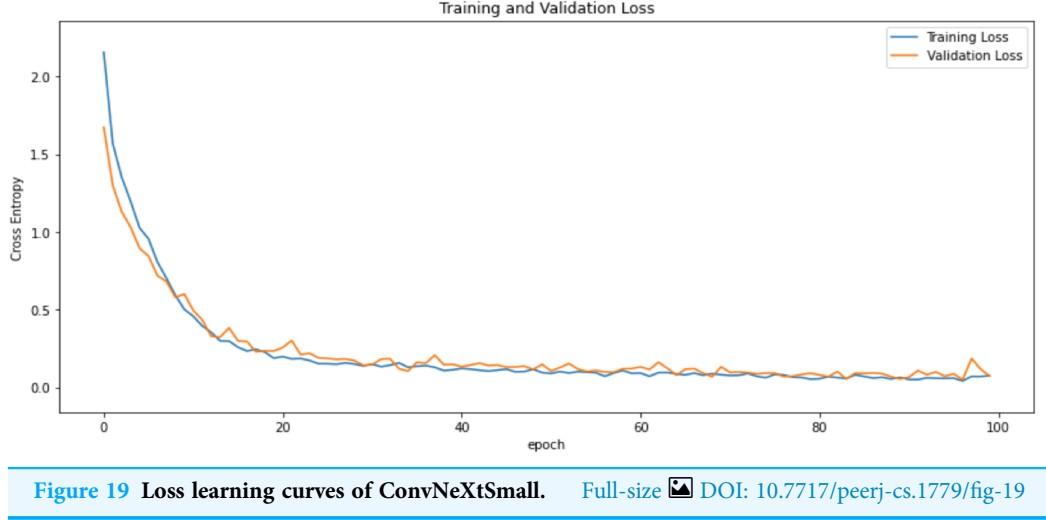

**Figure 19  Loss learning curves of ConvNeXtSmall.**

When all the results are considered, Xception can be considered to get the most desired outcomes with being the best or the second best in all metrics and the third in time performance. The VGG-16 based model got the results that can be considered the worst among its peers.

## DISCUSSION

As mentioned earlier, the Xception-based experimental model emerged as the top-performing model, achieving an impressive accuracy rate of 97.66%, which is widely recognized as a key metric of model success. In addition to exhibiting the lowest loss, the model also achieved the second-highest ROC AUC score, with a marginal difference of only 0.01%. Notably, high precision and high recall are typically challenging to achieve simultaneously, as these metrics are often at odds with each other. However, the Xception model achieved the highest values for both precision and recall, striking a remarkable

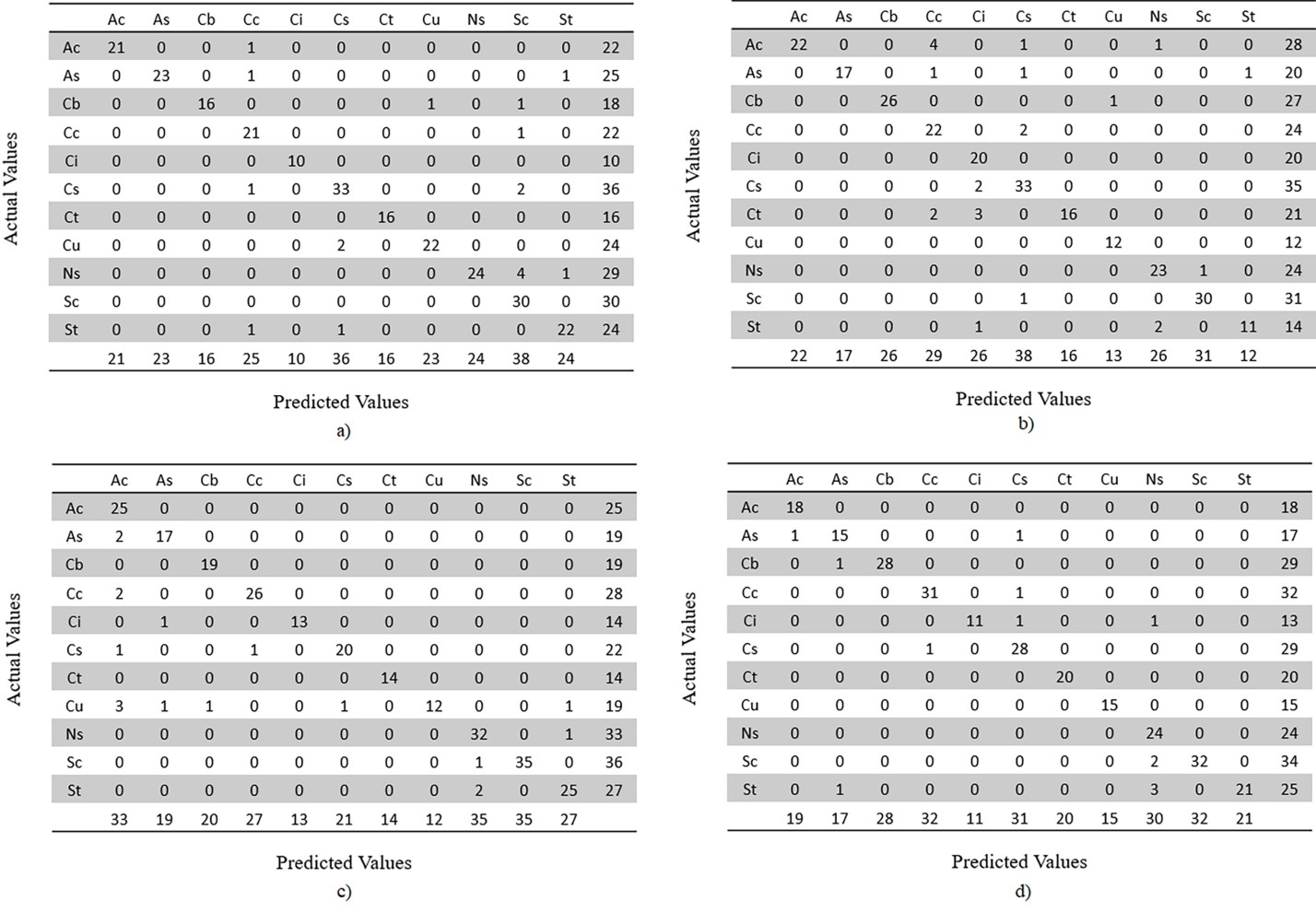

**Figure 20 Confusion matrices by classes, representing image counts of test dataset.** (A) MobileNet V2, (B) VGG-16, (C) ResNet-152 V2, (D) Inception V3.

balance represented by the F1-score. This underscores the effectiveness of leveraging the parallel convolution of InceptionNets for accurate cloud classification tasks.

The ConvNeXtSmall model indisputably secured the second position, demonstrating an accuracy of 97.27%, making it the second-best model with predictions surpassing 97%. Despite ranking fourth in precision, it exhibited the highest recall. However, given the emphasis on achieving balance in this experiment, the F1-Score provides a better indicator of the model's overall quality, revealing that the ConvNeXtSmall architecture delivered the second-best balance. Modernized convolutional neural networks (CNNs) aimed at bridging the gap with transformers also demonstrate promising capabilities in predicting such targets.

The EfficientNetV2L model can be considered the third most successful model, delivering satisfactory outcomes with an accuracy of 96.48%. In terms of precision-recall and their balance, this model leans toward precision, but the F1-score demonstrates a

**e)**

Actual Values / Predicted Values

|     | Ac | As | Cb | Cc | Ci | Cs | Ct | Cu | Ns | Sc | St |    |
|-----|----|----|----|----|----|----|----|----|----|----|----|----|
| Ac  | 26 | 0  | 0  | 0  | 0  | 0  | 0  | 0  | 0  | 0  | 0  | 26 |
| As  | 0  | 25 | 0  | 0  | 0  | 0  | 0  | 0  | 0  | 0  | 1  | 26 |
| Cb  | 0  | 0  | 17 | 0  | 0  | 0  | 0  | 0  | 0  | 0  | 0  | 17 |
| Cc  | 0  | 0  | 0  | 24 | 1  | 1  | 0  | 0  | 0  | 0  | 0  | 26 |
| Ci  | 0  | 0  | 0  | 0  | 15 | 0  | 0  | 0  | 0  | 0  | 0  | 15 |
| Cs  | 0  | 0  | 0  | 2  | 0  | 17 | 0  | 0  | 0  | 0  | 0  | 19 |
| Ct  | 0  | 0  | 0  | 0  | 0  | 0  | 21 | 0  | 0  | 0  | 0  | 21 |
| Cu  | 0  | 0  | 0  | 0  | 0  | 0  | 0  | 15 | 0  | 0  | 0  | 15 |
| Ns  | 0  | 0  | 0  | 1  | 0  | 0  | 0  | 0  | 25 | 0  | 3  | 29 |
| Sc  | 0  | 0  | 0  | 0  | 0  | 0  | 0  | 0  | 2  | 36 | 0  | 38 |
| St  | 0  | 0  | 0  | 0  | 0  | 0  | 0  | 0  | 0  | 0  | 24 | 24 |
|     | 26 | 25 | 17 | 27 | 16 | 18 | 21 | 15 | 27 | 36 | 28 |    |

**f)**

Actual Values / Predicted Values

|     | Ac | As | Cb | Cc | Ci | Cs | Ct | Cu | Ns | Sc | St |    |
|-----|----|----|----|----|----|----|----|----|----|----|----|----|
| Ac  | 28 | 0  | 0  | 0  | 0  | 0  | 0  | 0  | 0  | 0  | 0  | 28 |
| As  | 0  | 18 | 0  | 0  | 0  | 0  | 0  | 0  | 0  | 0  | 0  | 18 |
| Cb  | 0  | 0  | 22 | 0  | 0  | 0  | 0  | 0  | 0  | 0  | 0  | 22 |
| Cc  | 0  | 0  | 0  | 32 | 0  | 1  | 0  | 0  | 0  | 0  | 0  | 33 |
| Ci  | 0  | 0  | 0  | 0  | 15 | 0  | 0  | 0  | 0  | 0  | 1  | 16 |
| Cs  | 0  | 0  | 0  | 0  | 0  | 25 | 0  | 0  | 0  | 0  | 0  | 25 |
| Ct  | 0  | 0  | 0  | 0  | 0  | 0  | 23 | 0  | 0  | 0  | 0  | 23 |
| Cu  | 0  | 0  | 2  | 0  | 0  | 0  | 0  | 12 | 0  | 0  | 0  | 14 |
| Ns  | 0  | 0  | 0  | 0  | 0  | 0  | 0  | 0  | 26 | 0  | 3  | 29 |
| Sc  | 0  | 0  | 0  | 0  | 0  | 0  | 0  | 0  | 0  | 28 | 0  | 28 |
| St  | 0  | 0  | 0  | 0  | 0  | 0  | 0  | 0  | 6  | 0  | 14 | 20 |
|     | 28 | 18 | 24 | 32 | 15 | 26 | 23 | 12 | 32 | 28 | 18 |    |

**g)**

Actual Values / Predicted Values

|     | Ac | As | Cb | Cc | Ci | Cs | Ct | Cu | Ns | Sc | St |    |
|-----|----|----|----|----|----|----|----|----|----|----|----|----|
| Ac  | 30 | 0  | 0  | 1  | 0  | 0  | 0  | 0  | 0  | 0  | 0  | 31 |
| As  | 0  | 15 | 0  | 0  | 1  | 0  | 0  | 0  | 0  | 0  | 0  | 16 |
| Cb  | 0  | 0  | 22 | 0  | 0  | 0  | 0  | 0  | 0  | 0  | 0  | 22 |
| Cc  | 1  | 0  | 0  | 21 | 0  | 2  | 0  | 0  | 1  | 0  | 0  | 25 |
| Ci  | 0  | 0  | 0  | 0  | 15 | 1  | 0  | 0  | 0  | 0  | 0  | 16 |
| Cs  | 0  | 0  | 0  | 0  | 0  | 25 | 0  | 0  | 0  | 0  | 0  | 25 |
| Ct  | 0  | 0  | 0  | 0  | 0  | 0  | 20 | 0  | 0  | 0  | 0  | 20 |
| Cu  | 0  | 0  | 1  | 0  | 0  | 0  | 0  | 19 | 0  | 0  | 0  | 20 |
| Ns  | 0  | 0  | 0  | 0  | 0  | 0  | 0  | 0  | 30 | 0  | 0  | 30 |
| Sc  | 0  | 0  | 0  | 0  | 0  | 0  | 0  | 0  | 1  | 28 | 1  | 30 |
| St  | 0  | 1  | 0  | 0  | 1  | 0  | 0  | 0  | 1  | 0  | 18 | 21 |
|     | 31 | 16 | 23 | 22 | 17 | 28 | 20 | 19 | 33 | 28 | 19 |    |

**Figure 21 Confusion matrices by classes, representing image counts of test dataset.** (E) EfficientNetV2L, (F) Xception and (G) ConvNeXtSmall.

**Table 4 Evaluation metrics by base models.**

| Models | Accuracy (%) | Loss | Precision (%) | Recall (%) | F1-score (%) | AUC of ROC |
|--------|--------------|------|---------------|------------|--------------|------------|
| MobileNet V2   | 96.36 | 0.17 | 94.84 | 91.88 | 93.34 | 99.83 |
| VGG-16         | 86.33 | 0.42 | 87.42 | 84.78 | 86.09 | 98.75 |
| ResNet-152 V2  | 95.70 | 0.11 | 97.40 | 94.15 | 95.75 | 99.78 |
| Inception V3   | 95.31 | 0.10 | 96.92 | 93.55 | 95.21 | 99.91 |
| EfficentNetV2L | 96.48 | 0.10 | 97.51 | 95.86 | 96.68 | 99.95 |
| Xception       | 97.66 | 0.06 | 98.28 | 97.90 | 98.09 | 99.98 |
| ConvNeXtSmall  | 97.27 | 0.06 | 97.11 | 96.95 | 97.03 | 99.99 |

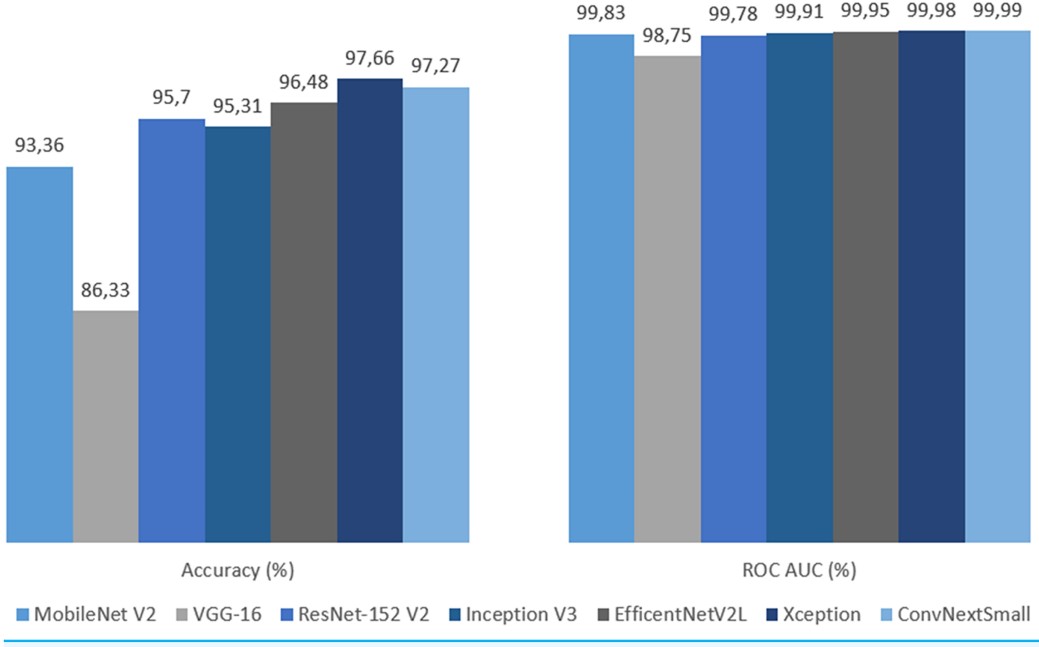

**Figure 22 Graph of accuracy and ROC AUC on test dataset.**

**Figure 23 Graph of precision, recall and F1-score on test dataset.**

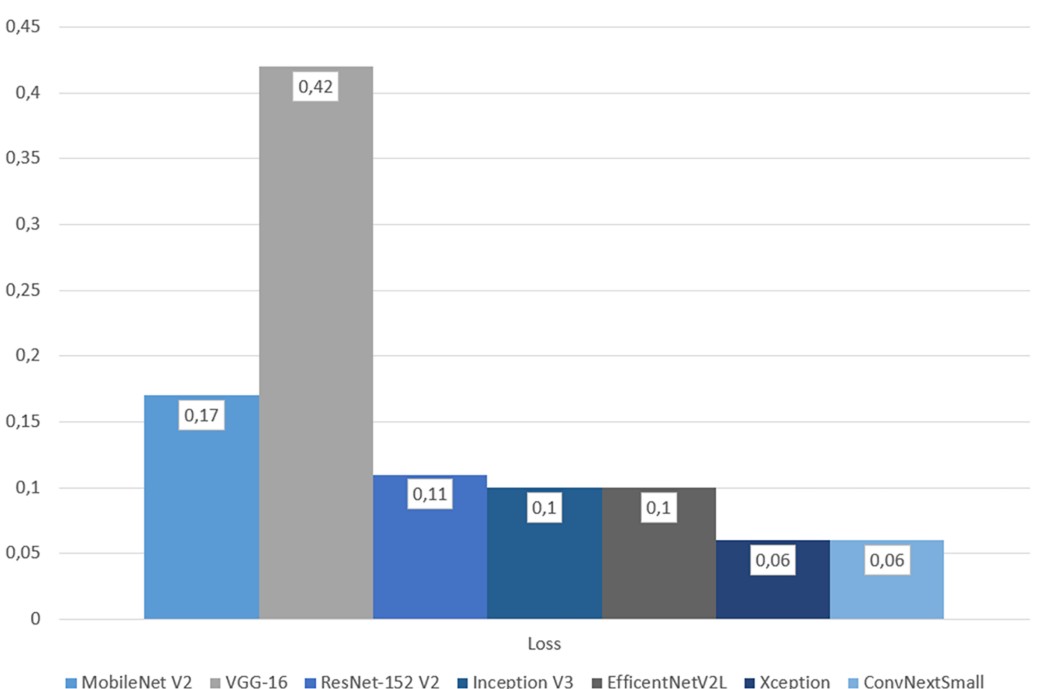

**Figure 24 Graph of categorical cross entropy loss on test dataset.**

**Table 5 Evaluation metrics by freezeout fine tuning.**

| Models | Accuracy (%) with fine tuning | Accuracy without fine tuning |
|---|---|---|
| MobileNet V2 | 96.36 | 95.24 |
| VGG-16 | 86.33 | 84.00 |
| ResNet-152 V2 | 95.70 | 94.86 |
| Inception V3 | 95.31 | 94.22 |
| EfficentNetV2L | 96.48 | 95.99 |
| Xception | 97.66 | 97.12 |
| ConvNeXtSmall | 97.27 | 96.59 |

**Table 6 Benchmark: comparison of the best accuracy in the latest technological approaches in the literature and this study.**

| Study | Best overall accuracy (%) |
|---|---|
| *Zhang et al. (2018)* | 88.8 |
| *Li et al. (2022)* | 92.70 |
| *Zhu et al. (2022)* | 95.60 |
| *Gyasi & Swarnalatha (2023)* | 97.45 |
| Ours (Xception) | 97.66 |

**Table 7 Training time table.**

| Models | Training time (min) |
|---|---|
| MobileNet V2 | 45 |
| VGG-16 | 68.33 |
| ResNet-152 V2 | 91.66 |
| Inception V3 | 56.66 |
| EfficentNetV2L | 145 |
| Xception | 65 |
| ConvNeXtSmall | 121.66 |

commendable balance between the two. Scaling the architecture's features to optimize efficiency can be argued as a successful approach for this challenge.

The primary model based on ResNet-152 V2 generated results that can be deemed intermediate among its peers. With an accuracy rate of 95.7%, ResNet-152 V2 achieved a high success rate in its evaluation metrics. Thus, very deep architectures incorporating residual mapping prove to be effective designs for this experiment.

MobileNet V2 is a model designed to prioritize speed while compromising prediction quality as minimally as possible. Although there are cases where this trade-off may result in a faster yet less accurate model, the 93.36% accuracy and other gratifying metric results validate MobileNet V2's proposed structure as a suitable choice in this experiment.

On the other hand, the VGG-16-based experimental model ranked last across all metrics and significantly lagged behind its counterparts. With an accuracy of 86.33%, it is the only model with accuracy below 90%. It exhibited the highest categorical cross-entropy loss, indicating a greater deviation from correct predictions compared to other models. Furthermore, it displayed the lowest precision and recall, along with the worst balance result, as reflected in the F1-score. Although VGG models are deep CNNs composed of small convolutional layers and have been successful in certain studies such as cloud coverage classification (*Kalkan et al., 2022*), VGG-16 produced disappointing results in this experiment.

In addition to evaluation metrics, another crucial factor for comparing the models is their time efficiency. As previously mentioned, training runtimes were measured for all models, providing the sole but sufficient data to assess this aspect. Unsurprisingly, the fastest model was MobileNet V2, with a runtime of 45 min, as it prioritizes excellent time performance. The Inception V3-based model secured the second position, closely followed by its "extreme version," Xception. Immediately after Xception, we have VGG-16, and these four models can be considered in the range of very fast to moderately fast. ResNet-152 V2 falls closer to the middle among all models. Lastly, ConvNeXtSmall ranks sixth, while EfficientNetV2L concludes as the slowest, both considered more resource-intensive models for data processing in this study.

These results clearly demonstrate that the Xception-based model not only produced the best results in terms of accuracy but also exhibited good time performance.

ConvNeXtSmall stands as the second-best model despite its slower architecture. MobileNet V2, EfficientNetV2L, ResNet-152 V2, and Inception V3 all possess the capability to deliver satisfyingly accurate predictions, with MobileNet V2 being the fastest, EfficientNetV2L the slowest, and the rest falling in between, leaning toward the faster side. However, VGG-16, despite its arguably time-efficient structure, generated the most unsatisfactory results.

When considering the entire set of experimental results, it can be confidently stated that the obtained outcomes are quite impressive, boasting very high accuracy and mostly acceptable performance in other aspects. In this regard, employing a CNN with transfer learning to classify cloud types proves to be an efficient approach, particularly when compared to one of the major inspirations for this research and the publisher of the dataset used in this experiment. The CloudNet achieved a maximum of 90% accuracy in class accuracies (*Zhang et al., 2018*), further highlighting the efficacy of the proposed approach.

## CONCLUSION

With the advancements in technology, accessing weather forecasts has become commonplace, especially with the widespread use of smartphones. However, despite its widespread usage, meteorological predictions still suffer from high error rates. One crucial factor that significantly influences weather forecasts is the accurate identification and classification of cloud types. Enhancing the precision of this process can help address this challenge. Moreover, the classification of clouds not only improves the accuracy of weather forecasts but also enables individuals to anticipate extreme weather events that have the potential to cause disasters. By leveraging cutting-edge artificial intelligence technologies, specifically deep learning, cloud type classification can be streamlined. Consequently, this study focuses on employing convolutional neural networks (CNN) with transfer learning, utilizing pre-trained models and fine-tuning techniques. The results obtained from this study are presented, compared, and interpreted herein.

Base models of MobileNet V2, Inception V3, EfficientNetV2L, VGG-16, Xception, ConvNeXtSmall, and ResNet-152 V2 were used, with the Xception model yielding the most promising outcome. It is worth noting that the accuracy of most models exceeded 95%, with the exception of VGG-16. These results far surpass previous works on the same subject, such as Cloud-MobileNet (*Gyasi & Swarnalatha, 2023*), which achieved the high accuracy of 97.45%. Hence, it can be concluded that transfer learning, coupled with freeze-out fine-tuning, has made a significant impact.

This academic study is anticipated to enhance existing meteorological forecast systems, improve predictions of extreme weather events, and serve as inspiration for future research in this field.

## APPENDIX

Figure A1 explains the workflow of the proposed approach which comprises pre-training, training, fine tuning, and lastly, evaluation phases.

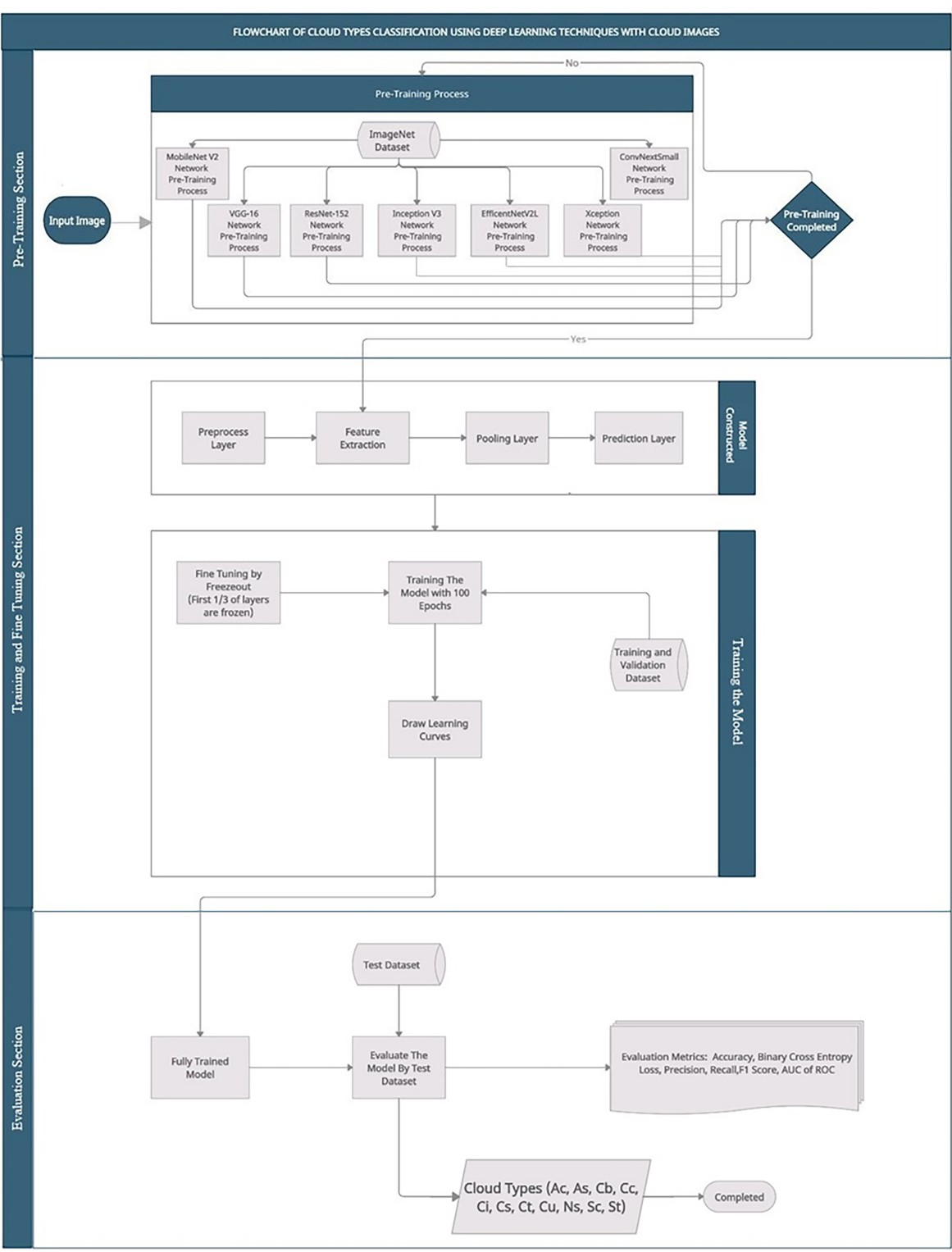

**Figure A1 Flowchart representation of the experiment.**

### Funding

The authors received no funding for this work.

### Competing Interests

The authors declare that they have no competing interests.

### Author Contributions

- Mehmet Guzel analyzed the data, performed the computation work, prepared figures and/or tables, and approved the final draft.
- Muruvvet Kalkan conceived and designed the experiments, performed the experiments, analyzed the data, performed the computation work, prepared figures and/or tables, and approved the final draft.
- Erkan Bostanci conceived and designed the experiments, performed the experiments, authored or reviewed drafts of the article, and approved the final draft.
- Koray Acici analyzed the data, authored or reviewed drafts of the article, and approved the final draft.
- Tunc Asuroglu analyzed the data, authored or reviewed drafts of the article, and approved the final draft.

### Data Availability

The data is available at Kaggle and Harvard Dataverse:

- https://www.kaggle.com/datasets/mmichelli/cirrus-cumulus-stratus-nimbus-ccsn-database.

- Liu, Pu, 2019, "Cirrus Cumulus Stratus Nimbus (CCSN) Database", https://doi.org/10.7910/DVN/CADDPD, Harvard Dataverse, V2.

The code is available at GitHub and Zenodo:

- https://github.com/mrsKalkan/GitHub-Categorical-ConvNeXtSmall-with-Confusion/tree/main.

- mrsKalkan. (2023). mrsKalkan/GitHub-Categorical-ConvNeXtSmall-with-Confusion: v1.0.0 (v1.0.0). Zenodo. https://doi.org/10.5281/zenodo.10141318.

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
