# Peer review of "Cloud type classification using deep learning with cloud images"

_PeerJ Computer Science, doi:10.7717/peerj-cs.1779_

## Round 0.1 · original submission · Major Revisions

Reviewers have suggested revision. You are required to revise your manuscript and address all the comments & suggestions of all the reviewers. Looking forward to receiving your revised manuscript.

**Language Note:** PeerJ staff have identified that the English language needs to be improved. When you prepare your next revision, please either (i) have a colleague who is proficient in English and familiar with the subject matter review your manuscript, or (ii) contact a professional editing service to review your manuscript. PeerJ can provide language editing services - you can contact us at copyediting@peerj.com for pricing (be sure to provide your manuscript number and title). – PeerJ Staff

·

Basic reporting

The article is well-written, employing professional and clear English throughout. Terminology specific to the domain of machine learning and cloud classification is used accurately. The introduction section provides a decent understanding of the significance of cloud classification. However, a deeper dive into prior literature, besides the mentioned CloudNet, would add more depth to the context. The manuscript appears to be self-contained, providing a clear narrative from the introduction of the problem to the evaluation of solutions. The results section robustly addresses the hypotheses, yet more detailed proofs or specifics about model configuration would be beneficial.

Experimental design

The research tackles the relevant and significant problem of cloud classification using state-of-the-art deep learning models, which seems to fit well within the broader scope of AI and meteorology. The article clearly defines its objective - enhancing cloud type classification for improved weather forecasting. The current state-of-the-art, CloudNet, serves as a reference, thereby indicating a knowledge gap which this study aims to fill. The use of multiple CNN models and a comprehensive evaluation indicates a technically rigorous approach. There's no explicit mention of ethical concerns or how they were addressed. The methods section could benefit from more explicit details on hyperparameters and configurations for each model, enhancing the study's replicability.

Validity of the findings

The study builds upon existing methods by applying a variety of CNNs to the cloud classification problem. While not entirely novel, it provides a comprehensive evaluation, which adds value to the literature. More comparative analysis with recent works would be a meaningful addition. The results showcase rigorous testing with statistical metrics. However, there is no explicit mention of the dataset's accessibility or if it has been shared in line with data sharing policies. The conclusions align well with the study's objectives. They provide a clear and concise summary of the findings, ensuring claims made are supported by the results.

Additional comments

Weaknesses:
1. Limited elaboration on the configurations and hyperparameters of the tested models. Clarify the exact configurations and hyperparameters used for each model to increase the study's replicability.
2. Absence of an in-depth comparative analysis with a wider range of related works or benchmarks, aside from CloudNet.
3. Enhance the methodology section by providing more specific details about the dataset: its size, the diversity of cloud types included, and the data distribution and Bias
4. Broaden the discussion by comparing the research findings with more recent studies or benchmarks in cloud type classification, in addition to the mentioned CloudNet.

for Example: you can compare yourself with : "https://github.com/marcosPlaza/Ground-based-Cloud-Classification-with-Deep-Learning"

Cite this review as

Reviewer 2 ·

Basic reporting

The paper is well written and focuses on an interesting topic. While the previous works included in the literature review section is not up to date, most of them are 5 or more years ago. It would be convincing to include more recent work.

All the figures are not high resolution, for example the title of each plot from figure 5-11 are invisible, and blur. This would be a huge rejection if not address properly.

Experimental design

Authors have conducted 7 DL approaches to compare. It would be more convincing to add shallow machine learning methods, since you compared training time in table 5.

Validity of the findings

The novelty of the study is limited, it is more like a technical report rather than a scientific study.

Cite this review as

Reviewer 3 ·

Basic reporting

1. This manuscripts anticipates cloud formations and classify them based on their shapes and colors, allowing for preemptive measures against potentially hazardous situation by leveraging AI techniques.

2. The paper is well-written.

3. Add more recent works, especially based on deep learning techniques.

4. Add a figure for main architecture of the proposed model.

5. Retain only one equation for F-score.

6. Check all typos and proofreading is needed.

Experimental design

- Add ablation study of the proposed model.

- Compare with at least one recent related work.

- create a new sub-section namely 'evaluation metrics' and place all metrics definitions and equation from 1 to 7.

- Create a table for hyper-parameter values for the proposed model (from line# 169-203).

Validity of the findings

Highlight best result in table 4 & 5.

Compare result with a recent state-of-the-art paper.

Additional comments

Major revision

Cite this review as

---

## Round 0.2 · accepted · Accept

The authors have addressed the comments and suggestions of all three reviewers. Manuscript may be accepted in the present form.

·

Basic reporting

no comment

Experimental design

no comment

Validity of the findings

no comment

Additional comments

All feedback has been addressed, and the article is now prepared for publication.

Cite this review as

Reviewer 2 ·

Basic reporting

My comments are all addressed. I have no further comments.

Experimental design

My comments are all addressed. I have no further comments.

Validity of the findings

My comments are all addressed. I have no further comments.

Cite this review as

Reviewer 3 ·

Basic reporting

Authors have addressed all comments. This manuscript can be acceptable in the present form.

Experimental design

Authors have addressed all experiments related comments.

Validity of the findings

Accept

Additional comments

Accept

Cite this review as